# CO-IMITATION LEARNING WITHOUT EXPERT DEMONSTRATION

**Kun-Peng Ning**[*]
ningkp@nuaa.edu.cn

**Hu Xu**[*]
xuhu1998@nuaa.edu.cn

**Kun Zhu**
zhukun@nuaa.edu.cn

**Sheng-Jun Huang**[†]
huangsj@nuaa.edu.cn

## ABSTRACT

Imitation learning is a primary approach to improve the efficiency of reinforcement learning by exploiting the expert demonstrations. However, in many real scenarios, obtaining expert demonstrations could be extremely expensive or even impossible. To overcome this challenge, in this paper, we propose a novel learning framework called *Co-Imitation Learning* (CoIL) to exploit the past good experiences of the agents themselves without expert demonstration. Specifically, we train two different agents via letting each of them alternately explore the environment and exploit the peer agent's experience. While the experiences could be valuable or misleading, we propose to estimate the potential utility of each piece of experience with the expected gain of the value function. Thus the agents can selectively imitate from each other by emphasizing the more useful experiences while filtering out noisy ones. Experimental results on various tasks show significant superiority of the proposed Co-Imitation Learning framework, validating that the agents can benefit from each other without external supervision.

## 1 INTRODUCTION

Reinforcement learning (RL) has achieved great success as a paradigm of learning intelligent agents for decision making. It aims to maximize the future reward received from environment by trading off exploration against exploitation Szepesvári (2010). To learn an effective policy, RL algorithms usually require a huge number of interactions with the environment, which is impractical in most real world applications Subramanian et al. (2016). To tackle this problem, imitation learning (IL) methods that follow the *learning from demonstrations* (LfD) framework try to reduce the number of interactions required for learning the policy by exploiting some expert demonstrations as the external supervised information Subramanian et al. (2016); Schaal (1997). However, in many tasks, the expert demonstrations could be extremely expensive or even impossible to obtain, such as in robotics Gabriel et al. (2019), self-driving cars Bojarski et al. (2017), finance Deng et al. (2016), or medical Miotto et al. (2018) applications. The high cost of expert demonstration makes the LfD methods less applicable.

Recently, an off-policy actor-critic algorithm called Self-Imitation learning (SIL) has been proposed to learn with the supervision from the agent's past good experiences Oh et al. (2018), instead of the expensive demonstrations. The key idea of SIL is to store past trajectories generated by the agent itself and allow the agent to imitate its past good action decisions. Although the SIL method does not rely on expert demonstration, its performance improvement could be less significant because the information contained in the past trajectories has been mostly utilized by the agent in previous training process.

Intuitively, the ceiling of performance could be broken if there is extra supervised information provided to guide the policy learning. Based on this motivation, we try to boost the agent by exploiting the experiences of another peer agent. Figure 1 shows an example to demonstrate our basic idea in

---

[*]Equal contribution
[†]Corresponding author.

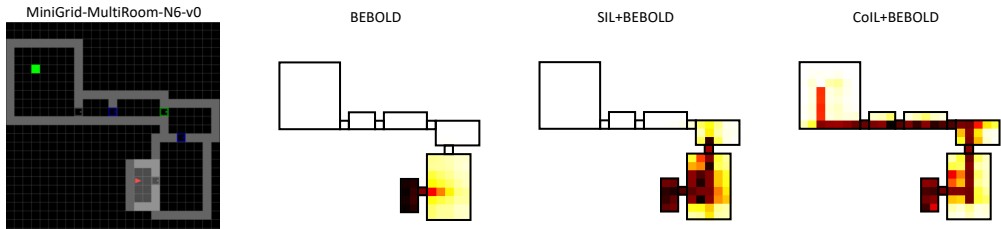

Figure 1: Heatmaps for the location of agents learnt with different algorithms in the *MiniGrid-MultiRoom-N6* environment at 1.5M training steps. The color depth represents the number of visitation counts for the agent's location. The co-imitation method can discover all the rooms and achieve the goal in 1.5M steps, while RL and SIL get stuck in the second and third room.

the *MiniGrid-MultiRoom-N6* task Chevalier-Boisvert et al. (2018b). In this reward-sparse environment, the agent across six rooms connected by doors and reach the green goal square. Intuitively, if we train two agents to explore different areas of the maze independently, and exchange their observed information, it is likely to be more effective for learning strategies than training only one agent. In other words, by imitating the good action decisions of peer agent in unknown states, the agent can give a better action than random to drive deeper exploration. To implement this idea, in this paper, we propose a novel framework called Co-Imitation Learning (CoIL). The CoIL framework trains two different agents to explore the environment, and lets them *imitate each other*. Specifically, we allow the two agents $\pi$ and $\hat{\pi}$ parameterized with $\theta$ and $\hat{\theta}$ to explore the environment separately, each of which will generate a series of trajectories. Then, to better exploit the peer agent's past experiences, we evaluate these experiences based on the difference between the cumulative value $\hat{R}$ calculated from peer one's trajectories and the state value $V_\theta(s)$ estimated by the state-value function of itself. This difference can be regarded as the expected gain from the peer agent's experience, and thus provides an estimation of its potential utility to policy learning. Based on the utility estimation, the agents then can selectively imitate from each other by emphasizing the more useful experiences with high utility while filtering out those with low utility. On one hand, by exploiting the peer agent's good experiences, the agent can learn a better potential policy to drive deep exploration. On the other hand, the experiences generated by exploring the environment can further be exploited by the peer agent to improve its performance.

We implemented our approach with multiple popular reinforcement learning algorithms, and perform experiments in different environments. The experimental results show that the policy learned by our CoIL approach achieves better performance in most cases, validating that the agents can benefit from each other without external supervision.

The rest of this paper is organized as follows. We review related work in the following section. Then the proposed approach is introduced. Next, experimental results are reported, followed by the conclusion.

## 2 RELATED WORK

To improve the efficiency of reinforcement learning, many approaches have attempted to learn from perfect expert demonstrations, such as DQfD Hester et al. (2018) and POfD Kang et al. (2018). The key idea of these methods is that RL algorithms can save many experiences by incorporating prior knowledge of various forms into the learning process. While the demonstrations could be noisy or imperfect, a few studies proposed to learn from imperfect demonstrations, such as LfND Ning & Huang (2020), NAC Gao et al. (2018) and so on. Different from these methods that require to utilize the expert demonstrations, our co-imitation learning aims to exploit the past good experiences of the agents themselves.

Recently, many approaches tried to improve learning in RL by exploring in a self-supervised manner. These methods study on how to construct intrinsic supervision in the learning process. The method in Pathak et al. (2019) trains an ensemble of dynamics models and incentivizes the agent to explore

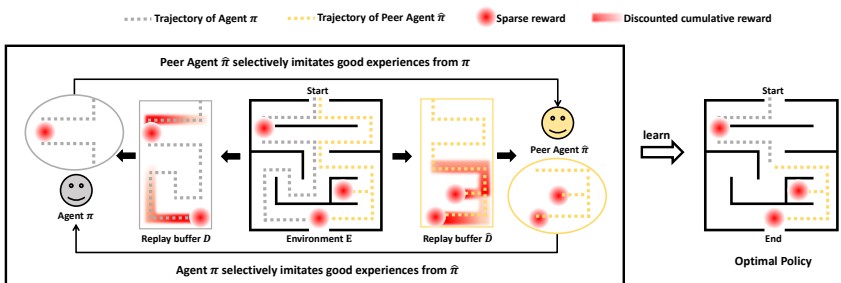

Figure 2: The Co-Imitation Learning (CoIL) framework. The trajectories of two agents include two sparse rewards respectively, and the optimal policy (including three sparse rewards) can be found effectively by imitating the actions with high potential utility.

such that the disagreement of those ensembles is maximized. The method in Pathak et al. (2017) tries to generate an intrinsic reward signal by self-supervised prediction, so as to make curiosity-driven exploration. Without external supervision, self-imitation learning Oh et al. (2018) attempts to reproduce the agent's past good decisions for deeper exploration. These methods usually achieve suboptimal performances without external supervision.

Our co-imitation learning approach is particularly effective in Hard-Exploration tasks with sparse rewards. Previous works make extensive use of intrinsic rewards as exploration bonuses, including count-based Bellemare et al. (2016); Burda et al. (2018), curiosity/surprise-driven Pathak et al. (2017) and state-diff approaches Zhang et al. (2019). Zhang et al. (2020) analyzes the pros and cons of the above method and proposes a better method called Beyond the Boundary of explored regions (BEBOLD), which achieves SoTA on various tasks. The key idea of these methods is to design intrinsic rewards to encourage exploration. Different from these methods, our approach studies how to exploit the past good experiences of the agents themselves.

## 3 THE CO-IMITATION LEARNING APPROACH

In this section, we begin with the background and the motivation, and then introduce the proposed Co-Imitation Learning framework in detail.

### 3.1 PRELIMINARIES

We consider an agent interacting with the environment over a sequence of steps, which can be formalized as a Markov Decision Process $\mathcal{M} = (\mathcal{S}, \mathcal{A}, \gamma, \mathcal{P}, \mathcal{R})$. Here, $\mathcal{S}$ is a set of states, $\mathcal{A}$ is a set of actions, $\gamma \in [0, 1)$ is the discount factor, $\mathcal{P}$ is the state transition probabilities, and $\mathcal{R} : (\mathcal{S} \times \mathcal{A}) \to \mathcal{R}$ is the reward function. At each step $t$, given the state $s_t \in \mathcal{S}$, the agent takes an action $a_t \in \mathcal{A}$ according to the policy $\pi$, and receives a reward $r_t \in \mathcal{R}$ from the environment. The target is to maximize the discounted sum of rewards over all step.

RL algorithms typically learn an effective control policy after many millions of interactions with the environment, and its learning process is rather inefficient. To overcome this deficiency, Self-Imitation Learning (SIL) Oh et al. (2018) stores past trajectories and allows the agent to self imitate its past good experiences to more effectively learn the policy $\pi$. Formally, the past episodes with cumulative rewards are stored in a replay buffer: $\mathcal{D} = \{(s_t, a_t, R_t)\}$, where $R_t = \sum_{k=t}^{\infty} \gamma^{k-t} r_k$ is the discounted sum of rewards. Then, it tries to exploit agent's past good experiences from the replay buffer to further improve the policy.

However, as discussed in the Introduction, SIL only attempts to mining the supervision from its own past experiences. Since there is no external information provided, the performance improvement could be less significant. In other words, SIL only emphasizes the past good experiences that the agent has utilized in previous training process, and thus limits its role in the environment exploration. Here we consider a "maze-search" problem as a supportive example. When there is only one agent to explore the maze, it is hard for the agent to walk out of the maze because it is easy to overestimate some observations by itself. Luckily, if we allow two agents to explore different areas of the maze

and let them exchange their observations, it will become much easier for them to find the exit of the maze. Intuitively, different agents can generate different action decision policies, and subsequently may generate experiences with different views. Based on this motivation, we try to train two agents by exploiting their experiences from two views to further improve the learning performance in RL.

## 3.2 ALGORITHM DETAIL

The framework of Co-Imitation Learning (CoIL) is demonstrated in Figure 2. It alternately explores the environment and exploits the peer agent's past experience in a joint framework, which allows the agents to effectively utilize the information from two sources. Specifically, in the first step, by exploring the environment, two agents will independently obtain a series of trajectories (dotted line) and sparse rewards (red circle), which will be stored in their replay buffers. Then, by estimating the potential utility of each piece of trajectory, the agents can selectively imitate each other from the peer agent's good experiences (black solid arrow) in the second step. As a result, the policies learned by cooperation can complement and reinforce each other, and thus can lead to a better performance and drive more effective exploration.

The goal of co-imitation learning (CoIL) is to train two agents to interact with the environment simultaneously, and let them imitate each other with peer agent's past good experiences. Without loss of generality, in addition to the current agent $\pi$, we introduce its peer agent as $\hat{\pi}$ parameterized by $\hat{\theta}$, and store its past experiences in a peer replay buffer: $\hat{\mathcal{D}} = \{(\hat{s}_t, \hat{a}_t, \hat{R}_t)\}$, where $\hat{s}_t$ is the state at time-step $t$, $\hat{a}_t$ is the action produced by $\hat{\pi}$, and $\hat{R}_t = \sum_{k=t}^{\infty} \gamma^{k-t} \hat{r}_k$ is the discounted sum of rewards.

Then, for agent $\pi$, to exploit the peer agent's past good experiences in the replay buffer $\hat{\mathcal{D}}$, we follow the loss form of Oh et al. (2018) and propose the following actor-critic loss $\mathcal{L}^{co}$ for CoIL.

$$\mathcal{L}^{co} = \mathbb{E}_{\hat{s}, \hat{a}, \hat{R} \in \hat{\mathcal{D}}}[\mathcal{L}^{co}_{policy} + \beta^{co} \mathcal{L}^{co}_{value}]. \tag{1}$$

The loss function consists of two parts, which are defined as follows respectively,

$$\mathcal{L}^{co}_{policy} = -log\pi_\theta(\hat{a}|\hat{s})(\hat{R} - V_\theta(\hat{s}))_+,$$

$$\mathcal{L}^{co}_{value} = \frac{1}{2}||(\hat{R} - V_\theta(\hat{s}))_+)||^2],$$

where $(\cdot)_+ = max(\cdot, 0)$, $V_\theta(\cdot)$ is the state value function of the current agent, and $\beta^{co}$ is the trade-off hyperparameter for the value loss.

It is worth noticing that we use peer agent's experiences in the replay buffer $\hat{D}$ to update the current agent $\pi$ by minimizing $\mathcal{L}^{co}$ in Eq 1. On one hand, by minimizing $\mathcal{L}^{co}_{policy}$, the policy $\pi$ will be optimized to produce consistent good action decisions with the peer agent. On the other hand, for an unseen state $\hat{s}$ in peer replay buffer $\hat{D}$, the state value function $V_\theta(\hat{s})$ will be optimized to have a more accurate estimation by minimizing $\mathcal{L}^{co}_{value}$.

Next, to make sure that the agents imitate the good experiences as much as possible but not the misleading ones, we propose to estimate the potential utility of each state-action pair based on the difference between the cumulative reward and the state value. Formally, given a state-action pair $(\hat{s}, \hat{a})$ from peer agent's experiences $\hat{D}$, its potential utility to learn $\pi$ is estimated as follows,

$$\mathcal{P}_\pi(\hat{a}, \hat{s}) = \hat{R} - V_\theta(\hat{s}). \tag{2}$$

Intuitively, if the action given by the peer agent $\hat{\pi}$ will lead to a higher cumulative reward than the current agent $\pi$, it is likely that this state has not been seen by the agent $\pi$ but has a high potential contribution to the policy training. Obviously, such experience should be imitated by the agent to improve the current policy network $\pi_\theta$. After imitating these useful experiences, when the current agent $\pi$ meets these states again, it is expected to take better actions according to these past experiences instead of random decisions.

Next, for environment exploring, we perform reinforcement learning algorithm to update two agents $\pi, \hat{\pi}$. Our co-imitation learning framework can be implemented with any actor-critic methods. In this paper, we follow the settings in Oh et al. (2018), and employ the advantage actor-critic (A2C) Frobeen (2017) and proximal policy optimization (PPO) as the basic algorithm of reinforcement learning. For convenience, the loss of RL is denoted as $\mathcal{L}^{rl}$.

---

**Algorithm 1** The Co-Imitation Learning algorithm

---

1: **Input:**
2:     Environment $\mathbf{E}$;
3:     Observation Space $\mathcal{O}$;
4:     Action Space $\mathcal{A}$;
5: **Process:**
6:     Initialize $\theta$ and $\hat{\theta}$ randomly;
7:     Initialize replay buffers $\mathcal{D} \leftarrow \emptyset, \hat{\mathcal{D}} \leftarrow \emptyset$;
8:     Initialize episode buffers $\mathcal{E} \leftarrow \emptyset, \hat{\mathcal{E}} \leftarrow \emptyset$;
9:     **for** each iteration **do**
10:         *# Collect experiences of two agents*
11:         The agent $\pi$ interacts with environment and its past experiences are stored in $\mathcal{E} = \{(s_t, a_t, r_t)\}$
12:         The peer agent $\hat{\pi}$ interacts with the environment and its past experiences are stored in $\hat{\mathcal{E}} = \{(\hat{s}_t, \hat{a}_t, \hat{r}_t)\}$
13:         *# Update replay buffers*
14:         Compute returns $R_t = \sum_{k=t}^{\infty} \gamma^{k-t} r_k$ for all $t$ in $\mathcal{E}$
15:         Compute returns $\hat{R}_t = \sum_{k=t}^{\infty} \gamma^{k-t} \hat{r_k}$ for all $t$ in $\hat{\mathcal{E}}$
16:         $\mathcal{D} \leftarrow \mathcal{D} \cup \{(s_t, a_t, R_t)\}$ for all $t$ in $\mathcal{E}$
17:         $\hat{\mathcal{D}} \leftarrow \hat{\mathcal{D}} \cup \{(\hat{s}_t, \hat{a}_t, \hat{R}_t)\}$ for all $t$ in $\hat{\mathcal{E}}$
18:         Clear episode buffers $\mathcal{E} \leftarrow \emptyset, \hat{\mathcal{E}} \leftarrow \emptyset$
19:         *# Perform reinforcement learning algorithm*
20:         $\theta \leftarrow \theta - \eta \nabla_\theta \mathcal{L}^{rl}$
21:         $\hat{\theta} \leftarrow \hat{\theta} - \eta \nabla_{\hat{\theta}} \mathcal{L}^{rl}$
22:         *# Perform co-imitation algorithm*
23:         **for** $m = 1$ to $M$ **do**
24:             Sample a mini-batch $\{(s, a, R)\}$ from $\mathcal{D}$
25:             $\hat{\theta} \leftarrow \hat{\theta} - \eta \nabla_{\hat{\theta}} \mathcal{L}^{co}$
26:             Sample a mini-batch $\{(\hat{s}, \hat{a}, \hat{R})\}$ from $\hat{\mathcal{D}}$
27:             $\theta \leftarrow \theta - \eta \nabla_\theta \mathcal{L}^{co}$
28:         **end for**
29:     **end for**
30: **Output:** the learned policies $\pi$ and $\hat{\pi}$.

---

The process of the proposed co-imitation learning (CoIL) approach is summarized in Algorithm 1. Firstly, the environment $\mathbf{E}$, observation space $\mathcal{O}$, and action space $\mathcal{A}$ are given. Then the parameters $\theta$ and $\hat{\theta}$ of the two agents are randomly initialized. We introduce two episode buffers $\mathcal{E}, \hat{\mathcal{E}}$ to store the two agent's past experiences, respectively, which are initialized as empty sets. Moreover, we also initialize two replay buffers $\mathcal{D}, \hat{\mathcal{D}}$ as an empty set for performing co-imitation. At each iteration, past experiences of two agents generated by interacting with environment are collected in the episode buffers $\mathcal{E}$ and $\hat{\mathcal{E}}$ respectively. Then, to update the replay buffers, we compute the cumulative rewards $R$ and $\hat{R}$ for all instances in $\mathcal{E}$ and $\hat{\mathcal{E}}$, and clear the episode buffers. After that, we perform reinforcement learning algorithm (minimizing $\mathcal{L}^{rl}$) and co-imitating (minimizing $\mathcal{L}^{co}$) from the replay buffers $M$ times to update the policy networks for each agent.

## 4 EXPERIMENTS

We conduct the experiments in continuous control MuJoCo Todorov et al. (2012) tasks and challenging procedurally-generate MiniGrid Chevalier-Boisvert et al. (2018a) environments. Specifically, for MuJoCo tasks, we perform experiments in four commonly used environments, namely *Swimmer*, *HalfCheetah*, *Ant* and *Walker2d*, which are implemented in OpenAI Gym Brockman et al. (2016). Moreover, we further conduct experiments by delaying reward the agent obtained from the environment, namely *DelayedSwimmer*, *DelayedHalfCheetah*, *DelayedAnt* and *DelayedWalker2d*, where the modified tasks give an accumulated reward after every 20 steps Oh et al. (2018). On

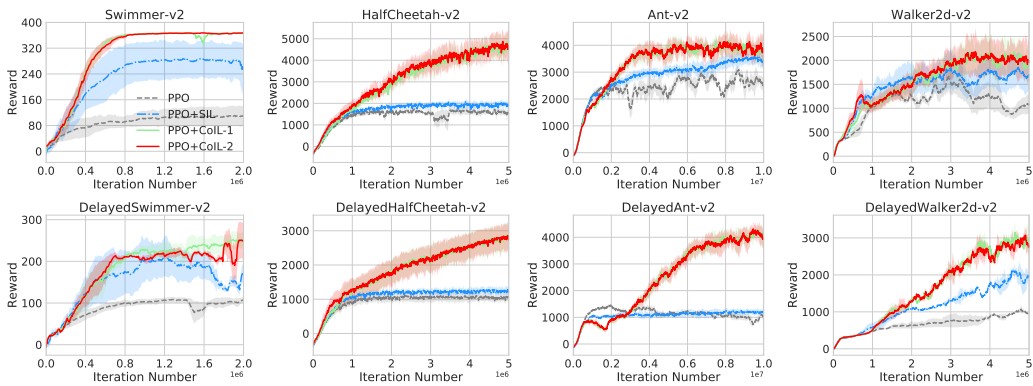

Figure 3: Results on OpenAI Gym MuJoCo tasks (top row) and delayed-reward versions of them (bottom row).

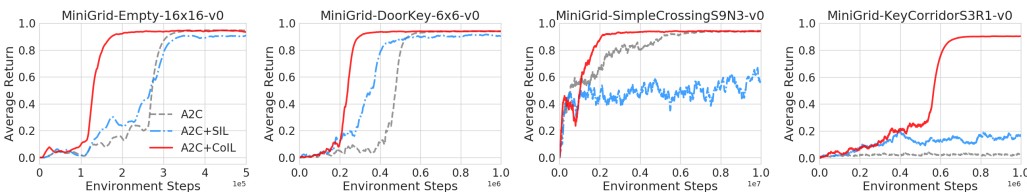

Figure 4: Results on four MiniGrid environments with sparse reward.

the other hand, for MiniGrid tasks, we perform experiments in *Empty-16×16*, *DoorKey-6×6*, *Key-CorridorS3R1* and *SimpleCrossingS9N3* environments, and four other relatively hard environments, namely *FourRooms*, *MultiRoom-N4*, *MultiRoom-N6* and *KeyCorridorS3R3*.

To validate the effectiveness of our approach, we compare the following methods in the experiments: $i$) **Proximal Policy Optimization (PPO):** a RL method that tries to optimize the lower bound of the clipped and unclipped objectives Schulman et al. (2017). $ii$) **Advantage Actor-Critic (A2C):** a RL method that uses advantage function as return in critic network Frobeen (2017). $iii$) **Beyond the Boundary of Explored Regions (BEBOLD):** the latest method proposes a new criterion for intrinsic reward to encourage exploration, which mitigates the short-sightedness issues in count-based methods and achieves a significant improvement over the previous SoTA. $iv$) **Self-Imitation Learning (SIL):** This method is designed to exploit the supervision from past good experiences of the agent itself. $v$) **Co-Imitation Learning (CoIL):** The approach proposed in this paper.

All the experiments are implemented in PyTorch Paszke et al. (2019) based on OpenAI's spinningup implementation Achiam (2018) and are conducted on four NVIDIA GTX 2080 Ti GPUs. For MuJoCo tasks, we used an MLP which consists of 2 hidden layers with 64 units as in Schulman et al. (2017). We performed 4 co-imitation learning as well as self-imitation learning updates per on-policy actor-critic update ($M = 4$ in Algorithm 1). For MiniGrid tasks, we used a 3-layer convolutional neural network with the image as input. We performed 10 co-imitation learning updates per each iteration (batch). Moreover, all the experiments are performed for 4 runs over 4 seeds (seed = 1, 2, 3, 4). More details of the network architectures and hyperparameters are described in the Appendix.

### 4.1 RESULTS ON MUJOCO

In this subsection, we evaluate the performance of compared methods by plotting the reward curve with the number of training iterations increases. The results with PPO as the base algorithm are demonstrated in Figure 3. We implement our CoIL approach base on PPO algorithm shown as **PPO+CoIL-$i$**, which denotes the performance of the $i$-th agent. Moreover, we also implement a combination with self-imitation learning shown as **PPO+SIL**.

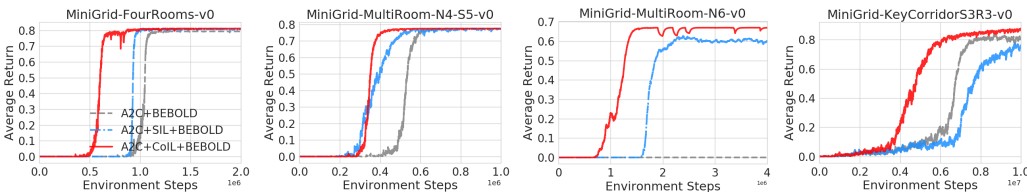

Figure 5: Results on four relatively hard MiniGrid environments.

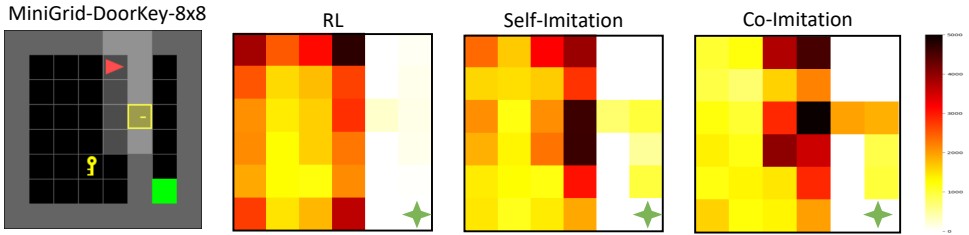

Figure 6: Heatmaps for the location of agents learnt with different algorithm in the MiniGrid-DoorKey environment at 800K training steps. The proposed co-imitation method can explore the right areas more efficiently by training two agents to learn from each other.

From Figure 3, we can observe that in both standard and delayed-reward environments, the proposed CoIL approach significantly outperforms the other methods in all cases. CoIL can achieve higher reward with fewer training iterations in general. When comparing with the PPO method that only explores the environment, the proposed CoIL method and the SIL method are always superior by utilizing the supervised information from past experiences. This phenomenon implies that it is more effective to learn policy by simultaneously exploiting the past experiences and exploring the environment. The SIL method, which aims to exploit past good experience of the agent itself, can improve learning based on PPO in some cases but fail to achieve higher performance in other cases (such as in *HalfCheetah*, *DelayedHalfCheetah*, and *DelayedAnt*). One possible reason is that SIL only exploits its own supervision without external information. In constract, the proposed CoIL approach, which allows the agent to obtain external supervision by imitating the peer agent's past good experiences, can complement and reinforce each other, and thus lead to a higher performance. Importantly, the proposed CoIL approach can significantly improve the performance of the Spinning Up benchmark Achiam (2018) implementation to a new level on all tasks. Especially in *HalfCheetah* environment, CoIL can get a reward of around 5000, while other methods can only get less than 2000. And in *Swimmer* environment, CoIL can achieve a reward of 360, which is significantly higher than the best results reported in previous literatures.

In standard MuJoCo environments, the reward structure is smooth and dense so that agent always receives a reasonable amount of reward and learns much faster in this type of domain. Thus, we further validate the robustness of our approach by delaying the reward the agent received from the environment. The result is demonstrated in the bottom row of Figure 3. In general, all compared methods for the delayed-reward tasks do not perform as well as those in the standard environments. However, it is clearly shown that the gap between CoIL and other methods is larger on delayed-reward tasks compared to standard tasks. More surprisingly, the CoIL methods can achieve competitive even higher performance in *DelayedAnt* and *DelayedWalker2d* environments. The results in both standard and delayed environments consistently validate the effectiveness of co-imitation learning, i.e., the agents can benefit from each other by imitating the peer agent's experience.

## 4.2 RESULTS ON MINIGRID

In this subsection, to examine the effectiveness of the proposed methods in sparse-reward environment, we follow the setting in Oh et al. (2018) to employ A2C as the base algorithm and perform experiments in various MiniGrid environments. Figure 4 demonstrates the performance comparison

in four environments, where **A2C+CoIL** and **A2C+SIL** respectively represents the combination of co-imitation learning and self-imitation learning with A2C.

From Figure 4, it can be observed that the proposed CoIL approach significantly outperforms the other methods in all cases. The policy learned by the proposed CoIL approach can find the goal with fewer training iterations. Especially CoIL can solve *KeyCorridorS3R1* task, while A2C and SIL both fail. It worth noting that the trajectory sequences generated on these tasks have strong dependence. For example, in the DoorKey domain, the agent needs to pick up the key, open the door and get to the goal, then achieves a sparse reward. Due to the sequential dependency for tasks, the policy learned by A2C has a low chance to find the goal, and thus leads to the slow convergence (such as in *Empty* and *DoorKey*) or even failure on some tasks (such as in *KeyCorridor*). The SIL method also fails in some environments due to similar reasons. Intuitively, if the current agent cannot find the goal or cannot produce good experiences, the SIL method will degenerate to standard A2C algorithm, and thus will not promote the learning ability to drive effective exploration. The proposed CoIL approach can effectively alleviate this issue. Intuitively, two agents have a greater chance to find the goal and generate good experiences. After exchanging past experiences and selectively imitating each other, two agents can benefit from each other's good decisions.

Moreover, to investigate whether our CoIL approach is complementary to count-based method (BE-BOLD), we further perform experiments on other four relatively hard environments. The results in Figure 5 show a consistent conclusion. When combined with BEBOLD approach, the exploration ability of all the compared methods are improved. It is clearly shown that the proposed approach still outperforms the other algorithms.This result also validates that co-imitation learning and count-based exploration can be well incorporated to solve the hard-exploration problems.

### 4.3 VISITATION COUNTS ANALYSIS IN MINIGRID

To study how different methods can effect the exploration of the agent, we further analyze the visualized results of visitation counts in *DoorKey* and *MultiRoom-N6* task. The target is to let the agent across the rooms and reach the green goal square. Moreover, we define visitation counts $N(s)$ at every state as the metric to measure the effectiveness of an exploration strategy. Figure 1 and 6 shows the heatmap of visitation counts at 1.5M and 800K steps respectively. It can be observed that the policy learned by our CoIL method can drive agent to explore more effectively. This visualized result also validate the superiority of the proposed co-imitation learning.

## 5 CONCLUSION

In this paper, we propose a new learning framework called Co-Imitation Learning (CoIL) to exploit the past good experiences of the agents themselves without expert demonstration. By training two different agents and letting each of them alternately explore the environment and exploit the peer agent's experience, the agents learned by cooperation can complement and reinforce each other. Moreover, with an adaptively strategy to estimate the potential utility of each piece of experience, the agents can selectively imitate from each other by emphasizing the more useful experiences. Experimental results on various tasks consistently demonstrate the superiority of the proposed CoIL approach. In the future, in order to further drive more effective exploration, we plan to emphasize the diversity between the two agents for producing more useful experiences.

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
