# OpenReview forum: "Co-Imitation Learning without Expert Demonstration"
_ICLR.cc/2023/Workshop/RRL — RRL 2023 Spotlight_

### Official Review · Reviewer_sMjn · 2023-02-20
**Promising approach for multi-agent reinforcement learning**

**Rating:** 4
**Confidence:** 4

**Review:**

# Summary
This paper presents a novel approach to reinforcement learning called Co-Imitation Learning. This enables two agents to share experiences and learn from each other without having access to expert demonstrations.

---

## Strengths
- The paper is organized and well-written
- The approach proposed in the paper is novel and interesting
- The authors compared CoIL to a strong multi-actor reinforcement learning algorithm like A2C

## Weaknesses
- The approach seems to be only applicable in situations where it is possible to place the agent in arbitrary locations in the environment, which may not be possible in many difficult reinforcement learning tasks.
- For large environments, a choice would have to be made about where the best places to start the different agents are.
- It is unclear how the utility factor is used in the loss function. $\mathcal{P}_{\pi}$ is not used anywhere in the paper (unless I missed it).

---

### Official Review · Reviewer_TLbj · 2023-03-01
**The more the merrier - faster and better imitation learning through cooperation between agents**

**Rating:** 4
**Confidence:** 5

**Review:**


This paper proposes a CoIL framework - Co-Imitation Learning, that trains two agents at the same time and “cooperatively” leverages past good experiences of the agents to learn a policy. This approach avoids the need for potentially expensive expert demonstrations in imitation learning, and is better than related approaches like self-imitation learning because the experiences of the other agent provide external supervised information.

The Introduction section explains the intuition behind the proposed idea in a simple and understandable manner using results from various approaches on the MiniGrid-MultiRoom-N6 task. The methodology's theoretical and practical aspects are clearly explained and motivated in subsequent sections. It is tested on various environments, and the selection of baselines and experimental setup is carefully documented. The results demonstrate the approach's applicability and effectiveness. Overall, the paper is succinct and makes for an interesting read.

The proposal is interesting and evokes similarities to bio-inspired algorithms like Particle Swarm Optimization. Here are a few suggestions on improvements and further work that would strengthen the paper but also serve as interesting future directions to pursue:
+ Extend the co-imitation learning approach to ensembles, i.e. more than 2 agents. It’d be interesting to contrast the improvements with the tradeoffs like greater resource requirements and training time per step.
+ More agents may lead to better generalization by avoiding local minimas. The phenomenon may be more evident with a higher number of agents and would be worth studying.
+ The paper mostly talks about cooperation between agents; it’d be interesting to explore the adversarial aspects of training multiple agents in tandem.